Myoferlin alleviates pressure overload-induced cardiac hypertrophy and dysfunction by inhibiting NLRP3-mediated pyroptosis

Zhou Yang 1
Liu Yanxu 1
Luo Hao 1
Wen Cong 1
Cui Yangyang 1
Du Linqing 1
Kwaku Ofe Eugene 1 2
Li Lan 1
Xiong Lijuan 3
Zheng Jiankang 1
Ding Xuefeng 4
Shen Xiufeng 1
Zhou Peng 3
Hu Houxiang 1
Yue Rongchuan yyc@nsmc.edu.cn 1 3
1 Department of Cardiology, Affiliated Hospital of North Sichuan Medical College , Nanchong , Sichuan , China
2 Family Health University College and Hospital, Opposite Kofi Annan International Peace Keeping Training Center , Teshie- Accra , Ghana
3 Department of Cardiology, People’s Hospital of Guang ’an District , Guang ’an , Sichuan , China
4 Department of Critical Care Medicine, Affiliated Hospital of North Sichuan Medical College , Nanchong , Sichuan , China
Haraguchi Tokuko
Electronic publication date: 2024 Nov 13
Publication date: 2024
Volume: 12
Electronic Location ID: e18499
Received 2024 Jun 28; Accepted 2024 Oct 18
Copyright: ©2024 Zhou et al.
Copyright year: 2024
Copyright holder: Zhou et al.
License: This is an open access article distributed under the terms of the Creative Commons Attribution License, which permits unrestricted use, distribution, reproduction and adaptation in any medium and for any purpose provided that it is properly attributed. For attribution, the original author(s), title, publication source (PeerJ) and either DOI or URL of the article must be cited.
License URL: https://creativecommons.org/licenses/by/4.0/

Keywords: Cardiac hypertrophy, Myoferlin, NOD-like receptor protein 3(NLRP3), Caspase 1 (CASP1)

Funding: The Research and development program of North Sichuan Medical College CBY23-TD01 Nanchong science and technology plan project 23JCYJPT0059 Sichuan Medical Science and Technology Innovation Research Society Program YCH-KY-YCZD2024-015 Sichuan Province Clinical Key Specialty Construction Project 2023GZZKP002 This study was supported by the Research and development program of North Sichuan Medical College (CBY23-TD01); Nanchong science and technology plan project (23JCYJPT0059), Sichuan Medical Science and Technology Innovation Research Society Program (YCH-KY-YCZD2024-015) and Sichuan Province Clinical Key Specialty Construction Project (2023GZZKP002). The funders had no role in study design, data collection and analysis, decision to publish, or preparation of the manuscript.

==============================
Myoferlin (MYOF) is a muscle-derived secretory protein. Recent studies have found that MYOF protects against cell damage. However, the role of MYOF in cardiac hypertrophy remains unclear. Increasing evidence suggests that NLRP3 (NOD-like receptor protein 3) and the pyroptosis cascade play critical roles in the development of cardiac hypertrophy and inflammation. To investigate the role of MYOF in cardiac hypertrophy, we conducted a transverse aortic constriction (TAC) experiment in a mouse model. We found that MYOF can improve cardiac hypertrophy and cardiac function. Furthermore, our study confirmed a connection between cardiac hypertrophy and myocardial pyroptosis. Cardiac hypertrophy significantly increased the proportion of apoptotic cells and upregulated apoptosis-associated speck-like protein containing a CARD (ASC), caspase-1, and gasdermin D (GSDMD). This suggests that pharmacological or genetic inhibition of NLRP3 can effectively reduce cardiac hypertrophy. An abnormal increase in NLRP3 can reverse the cardioprotective effects of MYOF. Our findings indicate that MYOF is a potential therapeutic agent for cardiac hypertrophy.

Introduction

Heart failure is the end stage of functional decline caused by pathological myocardial hypertrophy. Myocardial hypertrophy initially increases myocardial contractility. According to Laplace’s law, as the thickness of the left ventricle wall increases, the stress on the wall decreases, allowing the cardiac muscle to maintain its efficiency (Nakamura & Sadoshima, 2018). Cardiac hypertrophy progresses to the stage of decompensation and cardiac heart failure when pressure overload continues (Nakamura & Sadoshima, 2018; Oldfield, Duhamel & Dhalla, 2020). The risks associated with pathological myocardial hypertrophy include ventricular tachyarrhythmia, susceptibility to sudden cardiac death, heart failure, and other adverse cardiovascular events (Schiattarella & Hill, 2015). It is crucial to further understand the pathological mechanisms underlying cardiac hypertrophy to develop novel therapeutic options. By investigating these mechanisms, researchers can identify potential targets for intervention and develop novel treatment strategies to manage cardiac hypertrophy.

Unlike other forms of programmed cell death, such as apoptosis, pyroptosis is a pro-inflammatory form of cell death associated with the activation of caspase-1 (Bergsbaken, Fink & Cookson, 2009a). Inflammasomes are pattern recognition receptors for pathogens and other stimulated inflammatory responses. The NOD-like receptor protein 3 (NLRP3) inflammasome is a multiprotein complex comprising NLRP3, caspase-1, and ASC (apoptosis-associated speckle-like protein) (Yue et al., 2021). NLRP3 activation triggers the release of active caspase-1 (Yue et al., 2021), which in turn processes and releases the inflammatory cytokines IL-18 and IL-1β (Fink & Cookson, 2005). This process leads to the onset and spread of inflammatory responses. Pyroptosis appears late, and the NLRP3 inflammasome plays a vital role in cardiac remodeling (Wu et al., 2017). It is activated in several cardiac diseases, such as myocardial infarction, aortic valve disease, myocarditis, ischemia/reperfusion injury, hypertension, and atherosclerosis (Turner, 2016). In TAC (Transverse Aortic Constriction) mice, inhibition of NLRP3 has been found to reduce cardiac remodeling and enhance cardiac function (Li et al., 2017; Zhou et al., 2018), potentially, and further research is needed.

After exercise (Adams et al., 2007), MYOF, a muscle-derived secretory protein, showed a significant increase in expression. The increased MYOF expression after exercise suggests its importance in facilitating muscle development and growth in response to physical activity (Doherty et al., 2005). Cardiomyocytes are highly expressed in MYOF (Davis et al., 2000). The protective effect of MYOF against external stimulation-induced damage to cells has been demonstrated in several recent studies, mainly in the context of plasma membrane repair (Demonbreun et al., 2016). Elongated muscle contraction is prone to damage the plasma membrane and causes damage caused by contraction (Bernatchez et al., 2009). Therefore, the re-closure of the plasma membrane in muscle tissues is essential for the recovery from damage (Demonbreun et al., 2016). MYOF proteins contain multiple C2 structural domains located in the C-terminal transmembrane region. They have calcium-dependent phospholipid-binding activities that participate in calcium-associated cell membrane formation (Davis et al., 2000) and play roles in cell membrane repair (Bernatchez et al., 2009). Whether MYOF attenuates heart failure and myocardial hypertrophy by suppressing NLRP 3-mediated pyroptosis remains to be determined. The present study aimed to examine the impact of MYOF on pressure overload-induced myocardial hypertrophy and its influence on NLRP3-mediated pyroptosis in an in vivo model of myocardial hypertrophy.

Material and Methods

Main instruments and reagents

Surgical Microscope (Suzhou Liuliu Vision Technology Co. Ltd. China), Thoracic Spreader (Suzhou Medical Instrument Co. Ltd. China), Small Animal Ventilator (Chengdu Taimeng Software Co. Ltd. China), Biological Tissue Embedding Machine (Guangdong Saiwei Technology Co. Ltd. China), Tissue Slicer (Leica Corporation, Germany), Dehydrator (Hubei Huida Instrument Co. China), Pathological tissue bleaching and drying instrument (Changzhou HaoShiLin Medical Instrument Co. Ltd. China), and a constant-temperature water bath (Suzhou Medical Instrument Co. Ltd. China), laser scanning confocal microscopy (Olympus Inc., USA), 4′, 6-diamidino-2-phenylindole (4′, 6-diamidino-2-phenylindole, DAPI) staining solution (Biyuntian Institute of Biotechnology, China), NLRP3 monoclonal antibody (Santa Cruz Biotechnology Cat# sc-66846, RRID:AB_2152446), ASC monoclonal antibody (Santa Cruz Biotechnology Cat# sc-514414, RRID:AB_2737351), Caspase-1 monoclonal antibody (Novus Cat# NBP1-76606, RRID:AB_11022430), GSDMD monoclonal antibody (Abcam Cat# ab155233, RRID:AB_2736999), GAPDH antibody (Abcam Cat# 2251-1, RRID:AB_1267174), Wheat Germ Agglutinin Rabbit Polyclonal Antibody (Abcam Cat# ab20528, RRID:AB_445641), Tunel Kit (Roche, Switzerland), CRID3 (Cytokine Release Inhibitor Drug 3) (MCE, USA), Myoferlin (Phoenix Pharmaceuticals, USA), Tropomyosin Monoclonal Antibody (Abcam, USA), Cy3-labelled red donkey anti-goat IgG (Wuhan Aemetrix Technology Co. China).

Mouse model of myocardial hypertrophy

C57BL/6 mice were obtained from the Animal Center of North Sichuan Medical College (Nanchong, China). These animals were raised in an animal facility with five to six animals per cage, a constant temperature of 22 °C, relative humidity of 45%–55%, and a 12-h/12-h light/dark cycle. We used 4 month old male C57BL/6J mice to establish a model of pathological myocardial hypertrophy. Transverse aortic constriction (TAC) induced pathological myocardial hypertrophy due to cardiac pressure overload, according to a previously established method (Yue et al., 2021). The mice underwent anesthesia and were placed in a supine position on an operating table. An incision was made on the neck skin to expose the trachea, and the neck glands and muscles were dissected. A tracheal tube was then inserted to facilitate ventilator-assisted breathing. Subsequently, the thoracic cavity was opened to expose the aortic arch. A no. 7 nylon thread was threaded behind the first branch of the aorta. This procedure was performed to constrict the aorta to the thickness of a No. 27 syringe needle. The sham-operated control group was treated in a similar manner without ligating the aorta. The thoracic cavity was then sutured, and mechanical ventilation was maintained until spontaneous breathing was resumed. Following the completion of the surgery, the mice were provided with free access to water and food, which they could ingest as needed. At the end of the experiment, mice were euthanized with pentobarbital sodium (150 mg/kg) to reduce potential pain. Experiments were carried out in accordance with the animal protocols approved by North Sichuan Medical College Ethics Committee (NSMC Ethical Animal Trial [2023]086) and the ARRIVE guidelines. All methods were performed in accordance with relevant guidelines and regulations.

Experiment grouping

In order to determine the effect of MYOF on inhibiting pyroptosis and reducing pathological myocardial hypertrophy, mice were randomly divided into four groups: The control group (sham operation), MYOF group (mice undergoing sham operation received injections with MYOF via the tail vein at 2 mg/kg/week for 0, 1, 2, and 3 weeks postoperatively, The TAC group (mice undergoing TAC operation), and MYOF+TAC group (mice undergoing TAC operation were injected with MYOF via tail vein). To clarify whether myocardial hypertrophy during scorching was associated with NLRP3, mice were divided into four groups at random: the control group (undergoing sham surgery), the CRID3 group (receiving CRID3, 3mg/kg/week (Ward et al., 2019), via tail vein injections after sham surgery at 0, 1, 2, and 3 weeks postoperatively), the TAC group (undergoing TAC surgery), and the TAC+CRID3 group (undergoing TAC surgery and receiving tail vein injection of CRID3). In order to clarify whether the effect of MYOF on reducing pathological myocardial hypertrophy was related to NLRP3, mice were randomly assigned to the following four groups: control group (undergoing sham surgery), TAC group (undergoing TAC surgery), MYOF+TAC group (undergoing TAC surgery and injected with MYOF via the tail vein), and the pcDNA NLRP3+ MYOF+TAC group (injected with adeno-associated virus 9 (AAV9) vector carrying plasmid (pcDNA NLRP3 100µL vehicle containing 2.0 × 1011 AAV9 vector particles, intravenous injections) via the tail vein before TAC surgery and injected with MYOF via the tail vein after TAC surgery). Each group consisted of 12 mice. Myocardial tissues were acquired from the mice four weeks after surgery.

Echocardiography

To evaluate cardiac function, mice were anesthetized with a gas mixture of 2L/min oxygen inhalation and 2% isofluoroether, and at predetermined time intervals, mice underwent transthoracic echocardiography to evaluate cardiac function using GE VIVID 8D echocardiography. The heart was imaged on the short axis between the two papillary muscles, and the left ventricular ejection fraction (LVEF%) was determined by averaging the results of three consecutive heartbeats.

Immunohistochemistry

Mice were weighed and euthanized after 4 weeks, then hearts were removed and weighed, perfused with PBS, immobilized in 4% (w/v) paraformaldehyde, and embedded in paraffin. The tissue slices were incubated for 30 min at room temperature with goat serum. The purpose of this treatment was to prevent the binding of unspecific antibodies. Goat serum contains blocking agents that help reduce non-specific binding and minimize background staining, thereby improving the specificity and accuracy of antibody-based assays. After blocking, add the WGA solution, and incubate at 37 °C for 1 h. Wash the samples with PBS for 5 min each time, mount with a mounting medium, and observe under a microscope to quantify the cross-sectional size of the cardiomyocytes. The degree of fibrosis was evaluated by quantifying the entire Masson trichrome area of the left ventricular wall. A blinded quantification method was used to randomly select five high-magnification fields of view from each section. An auto-image analysis program was used to quantify the images in this study.

TUNEL assay

TUNEL detection was based on our previous methodology (Yue et al., 2021). In the study, cardiomyocytes were identified using myosin antibody staining specifically for myonectin. Additionally, nuclear staining was performed using DAPI. The images were captured using fluorescence microscopy. The mean fluorescence intensity was calculated using ImageJ software. Fifteen images were selected randomly, ensuring that they were not adjacent, for quantifying TUNEL-positive nuclei. The TUNEL-positive cell counts were analyzed in relation to the total number of nuclei identified via DAPI staining. This comparison was performed to determine the percentage of cells that exhibited TUNEL positivity. The assay was done in blinded replicates six times.

Western blotting

After each group of myocardial tissues was treated as appropriate, cellular proteins were extracted according to our previously established method (Brennan & Cookson, 2000). The protein concentration was quantified using the BCA kit. The protein samples were then loaded onto polyacrylamide gels for separation. Transfer the separated proteins to a PVDF membrane and block with 5% non-fat dry milk for 2 h. Next, incubate with different primary antibodies overnight at 4 °C. After washing with TBST, add the corresponding secondary antibodies and incubate at room temperature for 1 h. The PVDFs were cleaned again and then scanned using the ODYSSEY-IR gel scanning system to visualize the bands.

Statistical analysis

The data is expressed as the mean ± standard error of the mean (SEM). To ensure accuracy and reliability, each test was repeated at least six times. The primary data were analyzed using GraphPad Prism 5.0 software (San Diego, CA, USA). A one-way ANOVA test was performed to compare data between groups. Statistical significance was considered when the P-value was less than 0. 05 (P<0. 05).

Results

MYOF can reduce myocardial hypertrophy, fibrosis, and heart failure caused by overload stimulation

To clarify the contribution of MYOF to myocardial hypertrophy, we tested the impact of MYOF on overload stimulus-induced myocardial hypertrophy, fibrosis, and heart failure. An increased heart weight (HW)/body weight (BW) ratio following TAC suggests the development of myocardial hypertrophy due to excessive pressure overload (Figs. 1A and 1B). MYOF significantly inhibited the TAC-induced elevation of the HW/BW ratio, whereas MYOF itself did not alter the quality of healthy hearts. Wheat Germ Agglutinin (WGA) cross-sections were used for immunostaining to measure the cross-sectional area of cardiomyocytes. MYOF blunts the TAC-stimulated increase in the average surface area increase in cardiomyocytes (Figs. 1C and 1D). Pathological cardiac hypertrophy is a common cause of myocardial fibrosis and is a key factor in the development of heart failure. The extent of myocardial fibrosis in mice was detected by Masson’s staining four weeks after TAC. Significant interstitial fibrosis was observed in the TAC group, which was characterized by collagen accumulation (Fig. 1E). MYOF had no effect on myocardial fibrosis in sham-operated mice, but it significantly reduced the area of TAC-induced myocardial fibrosis (Fig. 1F). Echocardiography was used to examine the effect of MYOF on cardiac function in mice subjected to TAC-induced injury. It is shown in Fig. 1H that TAC resulted in insufficient cardiac function, as evidenced by reduced EF%, compared with the control group. MYOF significantly attenuated TAC-induced cardiac dysfunction but had no significant effect on cardiac function in sham-operated mice (Figs. 1G and 1H). MYOF was suggested to attenuate the deterioration of cardiac function and ventricular structure after TAC. These results suggest that MYOF attenuates overload-induced myocardial hypertrophy and heart failure.

Figure 1 MYOF reduces overload-induced cardiac hypertrophy.

(A) Representative images showing mouse hearts in the indicated groups. Scale bars: five mm. (B) Proportion of heart to body weight in the indicated groups. (C) Representative images showing the cross-sectional area of cardiomyocytes of the left ventricle in the indicated groups. Heart tissue were immune-stained with WGA. Green—anti-WGA. One hundred cells were randomly selected from five sections of each heart. Scale bars: 20µm. (D) Quantification of the cross-sectional area of cardiomyocytes of the left ventricle in the indicated groups. (E) Representative images showing myocardial fibrosis assessed using Masson trichrome staining in the indicated groups. The blue areas represent fibrotic staining. Fibrosis was quantified in five whole LV sections for each heart. Scale bars: 50µm. (F) Quantification of myocardial fibrosis in the indicated groups. The cardiac structure and function of mice were assessed by echocardiography 4 weeks after TAC. (G) Representative images of M-mode echocardiography of mouse hearts. (H) Quantitative analysis of left ventricular EF. Results are mean ± SEM, n = 6. * P < 0.01 compared to control, # P < 0.01 compared to TAC.

MYOF inhibits cardiomyocyte pyroptosis induced by overload stimulation

To investigate the correlation between pressure overload and pyroptosis, as well as the effect of MYOF on pyroptosis induced by pressure overload stimuli, the hearts of mice underwent various treatment regimens: control, MYOF, TAC, and MYOF + TAC. MYOF treatment also reduced TUNEL-positive cell counts compared with the TAC group (Figs. 2A and 2B). MYOF also reversed the TAC-induced increase in caspase-1 (Figs. 2C and 2D) and GSDMD (Figs. 2C and 2E) expression. These results indicate that MYOF plays a down-regulating role in pressure overload-induced cellular pyroptosis. These findings provide evidence that pyroptosis activation is associated with the development of pressure overload-induced cardiac hypertrophy.

Figure 2 MYOF inhibits pyroptosis induced by overload stimulation.

(A) Representative images showing TUNEL-positive cardiomyocytes. Red—tropomyosin, blue nuclei—DAPI, green nuclei—TUNEL. Scale bars: 100 µm. (B) Percentage of TUNEL-positive cells in the indicated groups. (C-E) Immunoblots showing the levels of caspase-1 (C and D) and GSDMD (C and E) in the indicated groups. Results are mean ± SEM, n = 6. * P < 0.01 compared to control, # P < 0.01 compared to TAC.

Inhibition of NLRP3 attenuates cardiomyocyte pyroptosis and cellular hypertrophy induced by overload stimulation

The expression of the NLRP3 inflammasome is upregulated in multiple inflammatory diseases. CRID3 (Cytokine Release Inhibitor Drug 3) is a selected inflammasome inhibitor of NLRP3. We further investigated the association between NLRP3 inflammasome activation and pyroptosis. The results of animal experiments showed that NLRP3 (Figs. 3A and 3B), caspase-1 (Figs. 3A and 3C), ASC (Figs. 3A and 3D), GSDMD (Figs. 3A and 3E) proteins, and following TAC treatment, there was an increase in the number of TUNEL-positive cells (Fig. 3F). It is worth noting that CRID3 reversed this increase. These data suggest that blocking the NLRP3 inflammasome with CRID3 inhibits TAC-induced cardiomyocyte pyroptosis. In this study, we investigated the effects of CRID3, an inhibitor of the NLRP3 inflammasome, on heart failure and myocardial hypertrophy. We found that treatment with CRID3 effectively suppressed the TAC-induced increase in the HW/BW ratio, whereas CRID3 itself did not alter the normal heart mass (Fig. 3G). Furthermore, our study demonstrated that CRID3 significantly prevented the increase in the mean surface area of cardiomyocytes induced by TAC (Figs. 3H and 3I). CRID3 had no effect on myocardial fibrosis in sham-operated mice but markedly decreased the area of TAC-induced myocardial fibrosis (Figs. 3J and 3K). CRID3 significantly improves cardiac function and reverses the decline in EF% stimulated by TAC (Figs. 3L and 3M). These data suggest that inhibition of the NLRP3 inflammasome by CRID3 reduces overload-stimulated myocardial hypertrophy and heart failure.

Figure 3 NLRP3 inhibitor CRID3 attenuates TAC-induced pyroptosis and myocardial hypertrophy.

(A-E) Immunoblots showing the protein levels of NLRP3 (A and B), caspase-1 (A and C), ASC (A and D), and GSDMD (A and E) in the cardiac tissue of indicated groups. (F) Percentage of TUNEL-positive cells in the indicated groups. (G) Proportion of heart to body weight in the indicated groups. (H-I) Representative images (H) and quantitative analyses of the cross-sectional area of left ventricular myocyte (I) in each group. (J-K) Representative images (J) and quantification of the myocardial fibrosis (K) in the indicated groups. (L-M) Representative images of M-mode echocardiograms of the heart (L) from each group and quantitative analysis of left ventricular ejection fraction (M). Results are presented as the mean ± SEM, n = 6. * P < 0.01 compared to control, # P < 0.01 compared to TAC.

Overexpression of NLRP3 reverses the inhibitory effect of myof on cardiomyocyte pyroptosis and cardiac hypertrophy

To evaluate the effect of exogenous MYOF on the control of NLRP3-stimulated inflammatory signaling, we generated adeno-associated viruses that overexpressed human NLRP3. We demonstrated that the empty vector had no effect on mice compared to the control group (Fig. S1). Western blot confirmed the successful overexpression of NLRP3 by pcDNA NLRP3 (Fig. S2). We evaluated the role of NLRP3 in anti-cardiomyocyte pyroptosis and myocardial hypertrophy in MYOF. MYOF-mediated inhibition of ASCs, caspase-1, and GSDMD (Figs. 4A–4E) was inhibited by high NLRP3 levels. Our experiments showed that compared with the MYOF+TAC group, NLRP3 overexpression can reverse the protective effects of MYOF, increase the proportion of apoptotic cells (Fig. 4F), the HW/BW ratio (Fig. 4G), cardiomyocyte cross-sectional area (Figs. 4H and 4I), myocardial fibrosis area (Figs. 4J and 4K), and weaken cardiac function (Figs. 4L and 4M). The data indicate that the anti-pyroptosis and cardiac hypertrophy effects of MYOF are associated with NLRP3 inflammatory signaling.

Figure 4 NLRP3 overexpression reverse MYOF-mediated antipyroptosis and anticardiac hypertrophy effects.

(A–E) Immunoblots showing protein levels of NLRP3 (A and B), caspase-1 (A and C), ASC (A and D), and GSDMD (A and E) in suitably treated cardiomyocytes. (F) Percentage of TUNEL-positive cells in suitably treated cardiomyocytes. (G) Proportion of heart to body weight in the indicated groups. (H–I) Representative images of myocardial cross-sectional area in each group (H). Quantification of the cross-sectional area of cardiomyocytes of the left ventricle in the indicated groups (I). (J–K) Representative images of Masson’s trichrome staining in each group (J). Quantification of the myocardial fibrosis in the indicated groups(K). (L–M) Representative images of M-mode echocardiograms of the heart from each group (L). Quantitative analysis of left ventricular EF in the indicated groups (M). Results are presented as the mean ± SEM, n = 6. P < 0.01 compared to control, # P < 0.01 compared to TAC.

Discussion

Our research demonstrated that MYOF has a beneficial effect in reducing TAC-induced cardiac hypertrophy. We also found that MYOF protected cardiomyocytes and cardiac function by suppressing NLRP3-induced pyroptosis. Finally, NLRP3 overexpression reversed the suppression of cardiomyocyte pyroptosis and hypertrophy induced by MYOF, suggesting that NLRP3-mediated pyroptosis is a major target of MYOF.

In simpler terms, pyroptosis is a type of programmed cellular self-destruct that is controlled by the caspase-1 (Ward et al., 2019). It is triggered by a series of microbial infections, such as Salmonella and Legionella infections. Furthermore, non-infectious stimuli (Bergsbaken, Fink & Cookson, 2009). The activation of caspase-1 degrades GSDMD, causing membrane pore formation and induction of pyroptosis. This process also leads to the emission of inflammatory factors (Ward et al., 2019; Brennan & Cookson, 2000; Luo et al., 2015; Bergsbaken, Fink & Cookson, 2009). Pyroptosis is a crucial factor in the development and progression of various cardiovascular diseases, such as atherosclerosis (Qian et al., 2021), ischemia/reperfusion injury (Liu et al., 2022), and heart failure (Habimana et al., 2022). The specific molecular pathways and triggers involved in pyroptosis-induced myocardial hypertrophy are still not fully understood. We established a mouse model of myocardial hypertrophy by performing TAC and noted a marked increase in pyroptosis. The increase in pyroptosis in myocardial hypertrophy was accompanied by enhanced regulation of NLRP3, caspase-1, ASC, and GSDMD. These results imply that myocardial pyroptosis is involved in the pathogenesis of cardiac hypertrophy.

MYOF is a protein of the berlin family of calcium/phospholipid-binding proteins with seven C2 structural domains (Han et al., 2019; Zhu et al., 2019). MYOF expression is high in skeletal muscle, cardiac muscle, and endothelial cells and plays a major role in vesicle transport, membrane fusion, and repair (Davis et al., 2000; Zhu et al., 2019; Posey Jr, Demonbreun & McNally, 2011). MYOF is abundant in myoblasts that are ready for fusion, and muscle injury triggers increased MYOF expression (Demonbreun et al., 2014; Doherty et al., 2008). Hence, MYOF mediates muscle damage in myocytes and myofibers (Doherty et al., 2005; Han et al., 2019). In our study, it was discovered that administering MYOF effectively reversed the elevated levels of NLRP3, caspase-1, ASC, and GSDMD that were abnormally observed after TAC treatment. The mouse model used in our study showed that treatment with MYOF resulted in a reduction in myocardial hypertrophy and fibrosis induced by TAC. This mainly manifested as a decrease in HW/BW, a smaller cardiomyocyte area, and improved EF, as suggested by echocardiography. Based on these findings, it can be concluded that MYOF can decrease myocardial hypertrophy by inhibiting pyroptosis.

Pyroptosis is triggered by rapid activation of inflammatory caspases, and the induction of pyroptosis depends on the presence of GSDMD, which is a protein targeted and activated by inflammatory caspases (Gaidt & Hornung, 2016). The N-terminal end of GSDMD forms a pore-like structure in the lipid membrane, resulting in the discharge of the inflammatory cytokines IL-18 and IL-1β (Sborgi et al., 2016). In response to tissue injury, both caspase-1-dependent and non-dependent pyroptosis can be triggered by inflammasome (Yue et al., 2021; Sborgi et al., 2016; Kovarova et al., 2012; Pronin et al., 2019; Xue et al., 2019; Yang et al., 2014). Our findings indicate that caspase-1-dependent pyroptosis contributes to the pathological processes associated with myocardial hypertrophy. CRID3 can target and inhibit the release of the NLRP3 inflammasome, reduce pyroptosis, attenuate pathological myocardial hypertrophy, and improve overall cardiac function.

Studies have shown that in a TAC-induced myocardial hypertrophy model, the NLRP3 inflammasome can be alleviated by regulating the NF- κB signaling pathway to alleviate myocardial hypertrophy (Ren et al., 2021). It has been proved that NF- κB may be the upstream signal of the NLRP3 inflammasome, and MYOF may alleviate myocardial hypertrophy by inhibiting the NF- κB signaling pathway. Research indicates that the depletion of MYOF can lead to mitochondria fission, with damaged mitochondria producing reactive oxygen species (Rademaker et al., 2018). Intracellular reactive oxygen species in cardiomyocytes are closely related to cardiac hypertrophy, and pathological cardiac hypertrophy can be inhibited through antioxidant stress therapy (Nakamura et al., 1998). By inhibiting Sirt4 in mitochondria, ROS production can be reduced, thereby alleviating pathological cardiac hypertrophy and fibrosis (Luo et al., 2017). MYOF may alleviate cardiac hypertrophy by influencing mitochondria to reduce ROS production. However, this is also a limitation of our experiment. In the future, we will continue to explore the upstream signaling pathways of MYOF to improve cardiac hypertrophy.

In the abovementioned study, we found that NLRP3-mediated pyroptosis has an essential pathological effect on the progression of myocardial hypertrophy. Additionally, MYOF mitigates pathological myocardial hypertrophy caused by pressure overload by inhibiting the proptosis cascade response. These findings suggest that MYOF can alleviate cardiac hypertrophy.

Conclusions

In this study, we established a mouse model of myocardial hypertrophy via TAC and investigated the pathological role of NLRP3 inflammasome-mediated cell pyroptosis, as well as the potential therapeutic effect of MYOF. Our experiments verified the protective effect of MYOF against TAC-induced pressure overload-induced cardiac hypertrophy. The mechanism of action may be associated with the NLRP3 inflammasome. The results of this study provide new avenues for treating myocardial hypertrophy.

Supplemental Information

Data S1 Raw data

Supplemental Information 2 Supplementary Figures

Supplemental Information 3 Western blot original material

Supplemental Information 4 Author Checklist

Additional Information and Declarations

Competing Interests

Author Contributions

Animal Ethics

Data Availability

The authors declare there are no competing interests.

Yang Zhou conceived and designed the experiments, performed the experiments, prepared figures and/or tables, and approved the final draft.

Yanxu Liu conceived and designed the experiments, performed the experiments, prepared figures and/or tables, and approved the final draft.

Hao Luo conceived and designed the experiments, performed the experiments, prepared figures and/or tables, and approved the final draft.

Cong Wen performed the experiments, prepared figures and/or tables, and approved the final draft.

Yangyang Cui analyzed the data, authored or reviewed drafts of the article, and approved the final draft.

Linqing Du analyzed the data, authored or reviewed drafts of the article, and approved the final draft.

Ofe Eugene Kwaku analyzed the data, authored or reviewed drafts of the article, and approved the final draft.

Lan Li analyzed the data, authored or reviewed drafts of the article, and approved the final draft.

Lijuan Xiong performed the experiments, analyzed the data, prepared figures and/or tables, and approved the final draft.

Jiankang Zheng analyzed the data, authored or reviewed drafts of the article, and approved the final draft.

Xuefeng Ding conceived and designed the experiments, analyzed the data, authored or reviewed drafts of the article, and approved the final draft.

Xiufeng Shen analyzed the data, authored or reviewed drafts of the article, and approved the final draft.

Peng Zhou performed the experiments, analyzed the data, authored or reviewed drafts of the article, and approved the final draft.

Houxiang Hu conceived and designed the experiments, authored or reviewed drafts of the article, and approved the final draft.

Rongchuan Yue conceived and designed the experiments, performed the experiments, prepared figures and/or tables, authored or reviewed drafts of the article, and approved the final draft.

The following information was supplied relating to ethical approvals (i.e., approving body and any reference numbers):

Experiments were carried out in accordance with the animal protocols approved by North Sichuan Medical College Ethics Committee (NSMC Ethical Animal Trial [2023] 086).

The following information was supplied regarding data availability:

The western blots are available in the Supplemental File.

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
