# Peer review of "Myoferlin alleviates pressure overload-induced cardiac hypertrophy and dysfunction by inhibiting NLRP3-mediated pyroptosis"

_PeerJ, doi:10.7717/peerj.18499_

## Round 0.1 · original submission · Major Revisions

The reviewers have found this manuscript valuable but also found some problems that need improvement. I think these opinions are reasonable. I hope that you will make revisions in accordance with the reviewers' opinions. In particular, please improve the points pointed out by reviewer 2, such as the figures being difficult to understand (the text in the figures is illegible).

Reviewer 1 ·

Basic reporting

no comment

Experimental design

This study established a mice model of myocardial hypertrophy through TAC and investigated the pathological role of NLRP3 inflammasome-mediated cell pyroptosis, as well as the potential therapeutic effect of myoferlin. This is a meaningful article, which provide the protective effect of myoferlin against pressure overload-induced cardiac hypertrophy induced by TAC. however, there were some major and minor issues to be addressed in the present manuscript.

Major Comments:
1. The logic of the abstract is unclear, the causal relationship is confused, and it is suggested to rewrite it;
2. Why not design the CRID3+ MYOF+TAC group in mice?
3. Introduction part: the NLRP3 was described as a protector in TAC mice, but in results, the level of NLRP3 was increased in TAC mice, they are contradictory;
4. What is the injection concentration and dose of adeno-associated virus 9 (AAV9) vector carrying plasmid (pcDNA NLRP3)? And immunofluorescence staining is preferred to demonstrate stable expression of AAV9- pcDNA NLRP3c;
5. Group AAV9-pcDNA blank needs to be designed as a control for group pcDNA NLRP3 to demonstrate that empty vectors have no effect on mice.

Minor Comments:
1. The first time a abbreviation appears there should be either a full name or a corresponding explanation to make it easier for the reader to understand the manuscript, GSDMD? MYOF? CRID3?
2. Paragraph format should be consistent throughout the manuscript.

Validity of the findings

no comment

Annotated reviews are not available for download in order to protect the identity of reviewers who chose to remain anonymous.

Reviewer 2 ·

Basic reporting

no comment

Experimental design

no comment

Validity of the findings

no comment

Additional comments

The manuscript entitled “Myoferlin alleviates pressure overload-induced cardiac hypertrophy and dysfunction by inhibiting NLRP3-mediated pyroptosis” studies the effect of Myoferlin in pressure overload-induced cardiac hypertrophy and dysfunction. The major findings are: a) Exogenous administration of Myoferlin attenuated TAC-induced pyroptosis and myocardial hypertrophy. B) Inhibition of NLRP3 with CRID3 attenuated TAC-induced pyroptosis, cardiac hypertrophy, and dysfunction. C) Overexpression of NLRP3 reversed myoferlin-mediated cardiac protection. My major concerns are as follows:
1. In the abstract (lines 27-28), the authors mentioned increased levels of IL-1β. I did not find any data showing IL-1β levels.
2. Figure 1 A1: The authors should put a scale bar on representative heart images.
3. Figure 1 B1: The representative WGA staining images are not clear. The authors should replace these images.
4. Figure 1D1: The representative echocardiographic images are not clear. Please replace the representative images with better quality images, and the representative images should reflect the quantified data of ejection fraction shown in Figure 1D2.
5. Figures 3 and 4: The author should improve the image quality. The font size of histogram labels should be increased.
6. Figure 4: The authors used adenovirus to overexpress NLRP3. The author should add a western blot image showing adenovirus injection increased NLRP3 expression (positive control).
7. The author should provide representative WGA images, fibrosis images, and echocardiographic images for Figures 3 and 4.
8. The authors injected 2µg/Kg/week Myoferlin and 3mg/Kg/week CRID3. Why did the author choose a particular concentration?

My minor concerns are:
1. The English language should be improved throughout the manuscript. For example: Line 124: (WGA)24. Lines 158-167: please rewrite this section. I suggest the authors revise the language in this manuscript to make it clearer to the audience.

---

## Round 0.2 · Minor Revisions

The reviewers have acknowledged that your paper has been improved. However, one of the reviewers has pointed out some minor issues. We would appreciate it if you could revise your manuscript accordingly.

Reviewer 2 ·

Basic reporting

no comment

Experimental design

no comment

Validity of the findings

no comment

Additional comments

The authors answered all of my comments.
Some minor comments to be addressed before accepting this manuscript:
1. Figure 2A label "MYOF+TA" should corrected to "MYOF+TAC".
2. Line 225 in the main manuscript: Shoddy-operated or sham-operated?
3. Line 237 in the main manuscript: myo+TAC group(?)
4. The authors should thoroughly review the manuscript to avoid any errors.
5. The authors have given supplemental figures 1 and 2 in response to the reviewers' documents. These supplementary figures should be described and cited in the result section, and separate supplementary materials documents should be prepared with these two figures.

---

## Round 0.3 · Minor Revisions

Your manuscript is almost at a stage where it can be accepted. However, there are some parts of the manuscript that need some corrections. For example, please make sure that the word "MYO" in line number 271 (word ms) is correct. If it is not correct, please correct it. There are many places in the manuscript where spaces are missing or needed. Please resolve these.

I look forward to receiving your revised manuscript.

---

## Round 0.4 · accepted · Accept

I am happy to accept this revised version of your paper.